# Parameter Identification of Fractional Index Viscoelastic Model for Vegetable-Fiber Reinforced Composite

**DOI:** 10.3390/polym14214634

**Published:** 2022-10-31

**Authors:** Angel Alexander Rodríguez Soto, José Luís Valín Rivera, Lavinia María Sanabio Alves Borges, Juan Enrique Palomares Ruiz

**Affiliations:** 1Escuela de Ingeniería Mecánica, Pontificia Universidad Católica de Valparaíso, Quilpue, Valparaíso 2430000, Chile; 2Faculdade de Engenheira Mecânica, Universidade Federal do Rio de Janeiro, Rio de Janeiro 21945-970, Brazil; 3Tecnológico Nacional de México/ITS de Cajeme, Cajeme 85024, Sonora, Mexico

**Keywords:** vegetable fiber-reinforced composite, viscoelasticity model, numerical method, material parameter determination

## Abstract

In the present work, parameters for adapting the behavior of the uniaxial three-element viscoelastic constitutive model with integer and fractional index derivatives to the mechanical evolution of an epoxy-composite material reinforced with long random henequen fibers, were determined. Cyclic loading–unloading with 0.1%, 0.2%, 0.3%, *…*, 1.0% controlled strain and staggered fluency experiments at 5 MPa, 10 MPa, and 15 MPa constant tension were performed in stages, and the obtained data were used to determine and validate the model’s parameter values. The Inverse Method of Identification was used to calculate the parameters, and the Particle Swarm Optimization (PSO) method was employed to achieve minimization of the error function. A comparison between the simulated uniaxial results and the experimental data is demonstrated graphically. There exists a strong dependence between properties of the composite and the fiber content (0 wt%, 9 wt%, 14 wt%, 22 wt%, and 28 wt% weight percentage fiber/matrix), and therefore also of the model parameter values. Both uniaxial models follow the viscoelastic behavior of the material and the fractional index version presents the best accuracy. The latter method was noted to be adequate for determination of the aforementioned constants using non-large experimental data and procedures that are easy to implement.

## 1. Introduction

In recent times, there has been a significant increase in the use of composite materials reinforced with vegetable fibers. Among these stand out the polymer matrix composites, with both biodegradable [1,2,3] and non-degradable polymers representing the most diversified among thermoset-based polymer composites [4,5,6]. The use of these fibers as reinforcement owe mainly to their lower cost, non-toxicity, low weight, and renewable nature [7,8]. One such composite material is an epoxy-matrix-composite reinforced with long randomly oriented Cuban henequen (sisal or agave fourcroydes) fibers, which presents the focus of this study.

For the design of parts and pieces composed of these materials, it is desirable to have digital models that can simulate their behavior over time under the action of external loads. These models, adjusted to the real behavior of the composites and having the ability to consider their “strain history”, can be implemented, for example, in Finite Element Method (FEM) analysis programs. As a basic first step, the unidimensional models can be generalized to two-dimensional and three-dimensional simulations. Therefore, it is necessary to determine the constants which adapt these models to the composite material behavior, achievable only through experimental data analysis.

Several authors have used fractional-order derivative index models to describe the mechanical behavior of viscoelastic materials, and experimental data to calculate the model constants. S. Müller [9] used experimental data to determine and validate the parameters of a fractional index model to characterize the mechanical behavior of thermoplastic polypropylene, performing one-dimensional and multi-dimensional simulations. H. Xu [10] defined the time-dependent creep behavior of several polymer materials and rock samples. This author uses the fractional Maxwell model, the fractional Kelvin–Voigt model, and the Poynting–Thomson model, and determined the model parameters by an interior-point method. When more factors need to be considered, such as temperature and the multi-shape memory effect of polymers, the multi-branch thermoviscoelastic fractional model, which uses fewer parameters, can achieve a more accurate approach to experimental data [11]. O. Martin [12] presented the quasi-static analysis of a simply supported viscoelastic beam subjected to uniformly distributed load, using the classical and the fractional Zener model. Abouelregal [13] describes the thermoviscoelastic behavior of rotating microbeams using the fractional Kelvin–Voigt model. Using fewer parameters to represent the experimental data and incorporating the Moore–Gibson–Thompson (MGT) equation of heat transfer, validation of the results was conducted through comparison with previous models and experimental values. This author observed that the fractional viscoelastic model better describes the damping properties than the classical integer differential-order model. A. Muliana [14] applied a numerical method to solve the nonlinear fractional viscoelastic constitutive model to simulate a time-dependent response of an isotropic polymer undergoing small deformation gradients. In this work, the Riemann–Liouville fractional integral for the time-dependent kernel function was considered and the material parameters were presented using experimental data. In this study, the proposed viscoelastic fractional model demonstrates the capability of describing multiaxial polymer responses under various loading histories with significantly fewer material parameters.

Diverse methods and strategies have been used for the calculation of the parameters. For example, Ruhani [15] used the statistical study of experimental data to infer a rheological model of a nanofluid with dynamic viscosity component. L. Cappelli [16] proposed a methodology based on the dynamic composite response to identify the viscoelastic behavior at the macroscopic and microscopic levels. These calculations take into account the minimum error between the numerical model and the experimental response of the material, using a general hybrid optimization tool that includes a genetic algorithm plus a gradient-based optimization approach. Z. Xiao [17] developed a numerical methodology to identify the constitutive parameters of the fractional Kelvin–Voigt model using an ACO (Ant Colony Optimization) algorithm to solve the inverse problem. W. Fan [18] estimated the parameters of the fractional element networks using the Bayesian statistical method, treating them as random variables with a distribution obtained from the experimental data. H. Xu [10] solved the nonlinear optimization problem through an interior-point algorithm, and determined the parameters of several fractional models. M. Shabani [19] recorded the viscoelastic deformation histories of silicon gel specimens through dynamic tests, and developed an identification strategy in the frequency domain, separating the real and imaginary parts of the frequency response function for error minimization. L. Viviani [20] modeled a laminated glass sandwich using a fractional approach, demonstrating the model’s effectiveness under blast loads, more convenient than the traditional Prony series approach. The fractional viscoelastic parameters were obtained via interpolation of the raw data.

The analytical resolution of differential equations with fractional index is difficult, and sometimes not possible. As seen in some of the works mentioned above, numerical methods are used to treat the reverse method of identification. Some of these methods are complex to implement and require the inclusion of several strategies and modifications to supply their deficiencies. Furthermore, for the calculation of parameters large amounts of data must be used, usually provided from complex experiments which incur high processing costs.

Nowadays, there are no models adjusted to the mechanical behavior of thermosettingpolymeric matrix composite materials reinforced with long random fibers of henequen, sisal, or similar materials. Motivated by the need to be able to simulate the temporal behavior of these materials under the action of external loads, the main objective of this work is to calculate the values of the parameters of the models that allow it. For this, we propose an innovative approach, selecting a rheological model with fractional index derivatives to address the viscoelastic behavior of these polymeric-matrix compounds reinforced with vegetable fibers. Hence, a smaller amount of experimental data is necessary for the adjustment of the model, compared to more complex models with integer index derivatives. Furthermore, it is necessary to highlight that the present work proposes a methodology that seeks to maintain simplicity and low computational cost while ensuring simulation capacity of the used model. For this reason, we used the Particle Swarm Optimization (PSO) method for the resolution of the Inverse Identification problem, through minimization of the error between the experimental data and those obtained by the models. For the calculation and validation of the parameters, we employed experimental data from simple cyclicincrementing loading–unloading tests and stepped fluency tests of brief time duration.

## 2. Material Model

Some authors have used rheological models that can simulate behavior of the material on one axis. After calculating the constants and validating them through comparisons with experimental data, this type of model can be taken to more complex levels. The mathematical representation of these materials includes the constitutive constants, stress, and deformation variables. Viscoelastic materials are strongly conditioned by the “strain history” and their performance depends on their rate. To produce more adequate models, capable of emulating realistic behavior, one of the researcher’s tendencies has been to increase its complexity and size via adding more elements. The connections, in series and/or in parallel, include elements that represent the elastic part and the viscose part of the model, elements that simulate non-linearities and equilibrium conditions [21]. This results in an increase in the number of the model parameters and therefore the need for a greater amount of experimental data. These are more difficult to analyze, incur high computational cost, greater processing time, and increase the risk of obtaining combinations of values that satisfy the imposed criteria but that are not correct.

To justify the use of fractional operators in viscoelastic material models, the approach was verified to be coherent with molecular theories [22] and are thermodynamically consistent [23]. These types of models require fewer constants for their adaptation to the real behavior of the material, comparing them with more complex models with entire differential indices [24]. It has been demonstrated that these models naturally include “memory effects”, allowing time-dependent models to be obtained [25]. To mention a few examples, A. Mohamed [26] simulated the mechanical performance CNTs/PES and graphene/PES membranes with different concentrations of nanofillers using a nonlinear uniaxial model based on the generalized Maxwell model. N. Vaiana [27] proposed uniaxial asymmetric models for simulating the mechanical hysteresis phenomena. K.F. Li [28] studied the behavior of PVA-ECC composites using long-term compressive creep experimental data and the Maxwell uniaxial model with fractional and integer order. U. Hofer [29] used the fractional Zener model and the extended Lomnitz model as a base to produce multiscale modeling of braid-reinforced polymers, which were subsequently validated through experimental data.

The linear standard one-dimensional viscoelastic model with three elements is one of the most commonly used approaches. Replacing the viscous element with a fractional index element can be achieved in the fractional Zener model [30], Figure 1, the stress–strain relation between these two models is shown in Equations (1) and (2). In addition, it was shown that this model is “quantitatively equivalent, under various loading conditions, to the generalized Maxwell model” [31].
(1)$$\frac{\sigma }{E}+\frac{\eta }{EE_{1}}\cdot \dot{\sigma }=\varepsilon +\left(\frac{\eta }{E_{1}}+\frac{\eta }{E}\right)\cdot \dot{\varepsilon }$$
(2)$$\frac{\sigma }{E}+\frac{\eta }{EE_{1}}\cdot \frac{d^{\alpha }\sigma \left(t\right)}{dt^{\alpha }}=\varepsilon +\left(\frac{\eta }{E_{1}}+\frac{\eta }{E}\right)\cdot \frac{d^{\alpha }\varepsilon \left(t\right)}{dt^{\alpha }}$$
where the constants of the modulus of elasticity and viscosity, and the index of the derivative, *E*, $E_{1}$, η and α, are intrinsic to the material, and $d^{\alpha }\left(\right)/dt^{\alpha }$ is the left-side Riemann–Loiuville differential operator of index (0<α<1), defined on the time interval, starting from zero, [0;b] similar to ${}_{0}D_{t}^{\alpha }\left(\right)$ [32].

Furthermore, the effective global properties of the composite were used as an equivalent homogeneous material [33], and assumed a uniform distribution and a random orientation of the reinforcing fibers. In addition, possible agglomerations of the reinforcements are not considered, allowing for assuming global isotropic properties in the plane of the loads.

## 3. Numerical Methodology

The present work used the methodology proposed by I. Podlubny [32], which uses triangular-strip matrices for discretization of the differential equation with an arbitrary real index, Equation (5). This methodology allows a simple implementation in specialized software. For this study, we used MatLab and a toolbox published by Podlubny [34].

In summary, the left-sided Riemann–Liouville derivative $d^{\alpha }\left(\right)/dt^{\alpha }$ or ${}_{0}D_{t}^{\alpha }\left(\right)$, Equation (3), using the backward fractional difference approximation can be represented in matrix form, Equation (4), using the triangular-strip matrix, Equation (5), where $B_{N}^{\alpha }$ is the analogue of the fractional derivative operator. For the study case, the time domain is discretized in N+1 equidistant points, with $t_{0}=0$ and $t_{N}=b$ and 0<α<1.
(3)$${}_{0}D_{t}^{\alpha }f\left(t\right)=\frac{1}{\Gamma (m-\alpha )}\frac{d^{m}}{dt^{m}}\int_{0}^{t}\frac{f(\tau )d\tau }{{\left(t-\tau \right)}^{\alpha +1-m}}$$
(4)$$\left[\begin{array}{c} h^{-\alpha }\nabla^{\alpha }f\left(t_{0}\right)\\ h^{-\alpha }\nabla^{\alpha }f\left(t_{1}\right)\\ h^{-\alpha }\nabla^{\alpha }f\left(t_{2}\right)\\ on\vdots \\ h^{-\alpha }\nabla^{\alpha }f\left(t_{N-1}\right)\\ h^{-\alpha }\nabla^{\alpha }f\left(t_{N}\right) \end{array}\right]=B_{N}^{\alpha }\left[\begin{array}{c} f_{0}\\ f_{1}\\ f_{2}\\ \vdots \\ f_{N-1}\\ f_{N} \end{array}\right]$$
(5)$$B_{N}^{\alpha }=\frac{1}{h^{\alpha }}\left[\begin{array}{cccccc} w_{0}^{\left(\alpha \right)} & w_{1}^{\left(\alpha \right)} & w_{2}^{\left(\alpha \right)} & \ddots & w_{N-1}^{\left(\alpha \right)} & w_{N}^{\left(\alpha \right)}\\ 0 & w_{0}^{\left(\alpha \right)} & w_{1}^{\left(\alpha \right)} & w_{2}^{\left(\alpha \right)} & \ddots & w_{N-1}^{\left(\alpha \right)}\\ 0 & 0 & w_{0}^{\left(\alpha \right)} & w_{1}^{\left(\alpha \right)} & \ddots & w_{N-2}^{\left(\alpha \right)}\\ \vdots & \ddots & \ddots & \ddots & \ddots & \vdots \\ 0 & \ddots & 0 & 0 & w_{0}^{\left(\alpha \right)} & w_{1}^{\left(\alpha \right)}\\ 0 & 0 & \cdots & 0 & 0 & w_{0}^{\left(\alpha \right)} \end{array}\right]$$

The Inverse Method of Identification was followed, where it is known that the effects (experimental data) can be determined by the causes (the material constants that adjust the model to the real behavior) [35]. For that reason, an iterative search of the model parameters was carried out that minimizes the differences (D) between the experimental data and the model response, where Equation (6) is the objective equation.
(6)$$D=\sum \sqrt{{\left[y_{r}^{exp}-y_{r}^{mod}\right]}^{2}}$$
where $y_{r}^{exp}$ was the *r* component of the experimental data vector and $y_{r}^{mod}$ was the *r* component of the equivalent model response vector. These vectors alternate between stress and strain data, depending on the entrance signal for the experiments.

In the iterative search process, the Particle Swarm Optimization (PSO) method was used. Some of this method’s advantages are that it has fewer parameters to adjust, and its simplicity makes it easy to implement and combine with other optimization strategies. This stochastic method imitates the behavior of large populations of individuals, moving through the “hyperspace of the possible solutions”, evaluating the input data as the coordinates of their position in the objective Equation (fitness) and sharing information of the best results with each other [36].

The *n*-dimensional search space of the possible solutions $\left(S\subset R^{n}\right)$ are limited by the vector of maximums $\left(b_{sup}\right)$ and the vector of minimums $\left(b_{inf}\right)$ of the searched model parameters. Similar to other population-based optimization algorithms, PSO stars with random positions and velocities at the beginning of all particles, see Equation (7).
(7)$$x_{i}^{0}∼U\left(b_{inf},b_{sup}\right);\;v_{1}^{0}∼U(|b_{sup}-b_{inf}\left|,-\right|b_{sup}-b_{inf}\left|\right)$$
where $x_{1}^{0}$ and $v_{i}^{0}$ are the position and velocity of the *i*-particle in the initial iteration, and U(α,β) is a random function limited by the vectors α and β.

The “movement” or position update $x_{i}^{k+1}$, see Equation (9), of each *i*-particle is governed by a velocity vector $v_{i}^{k+1}$, see Equation (8). This vector takes into account the velocity $v_{i}^{k}$ and the position $x_{i}^{k}$ of each particle in the previous *k*-iteration. Furthermore, taken into account are the constants that govern the “iteration inertia” $w^{k}$, the “weight” of the above information $C_{1}$ and $C_{2}$ and random constants $R_{1},R_{2}∼U\left[1;0\right]$, generated new for each iteration. The velocity vector of the particles that were out of the mentioned limits was inverted and divided by two.
(8)$$v_{i}^{k+1}=w^{k}\cdot v_{i}^{k}+C_{1}\cdot R_{1}\cdot \left(p_{i}^{k}-x_{1}^{k}\right)+C_{2}\cdot R_{2}\cdot \left(p_{glob}^{k}-x_{i}^{k}\right)$$
(9)$$x_{i}^{k+1}=x_{i}^{k}+v_{i}^{k+1}$$

At the end of each iteration, the objective function, Equation (6), was evaluated as f(), and the positions with the best global $p_{glob}^{k}$ and individual $p_{i}^{k}$ results are updated, Equations (10) and (11).
(10)$$if\;f\left(x_{i}^{k+1}\right)<f\left(p_{i}^{k}\right)\;then\;p_{i}^{k}=x_{i}^{k+1}$$
(11)$$if\;min\left[f\left(x_{i}^{k+1}\right)\right]<f\left(p_{glob}^{k}\right)\;then\;p_{glob}^{k+1}=x_{i\to (min)}^{k+1}$$

The balance between global and local searching throughout the process is critical to the performance of the optimization algorithm. To obtain an equilibrium between “exploration” and “exploitation”, the inertia weight coefficient was introduced *w* [37], a large value increases the exploration and a small value the exploitation. The diminution of the inertia *w* was defined in a homogeneous distribution across the iteration increase *k*, see Equation (12), previously, the maximum and minimum of the values, $w_{max},w_{min}$, were established.
(12)$$w^{k}=\frac{\left(w_{max}-w_{min}\right)\cdot \left(k_{max}-k\right)}{k_{max}}+w_{min}$$

Two stop criteria, the maximum quantities of iteration $k_{max}$ and the minimum relative error ea were introduced, Equation (13), ensuring that the iterations end and allow comparison with a preset relative error.
(13)$$ea=\left|\frac{f\left(p_{glob}^{k+1}\right)-f\left(p_{glob}^{k}\right)}{f\left(p_{glob}^{k}\right)}\right|$$

## 4. Materials and Methods

The general scheme of the work, shown in Figure 2, follows a similar strategy to some of the aforementioned works. First, experimental samples of composite materials with different fiber proportions were obtained. Subsequently, the load–unload and staggered creep experiments were performed. The viscoelastic models, of integer-order and fractional-order derivative index, were implemented in a MatLab program. Through an iterative process of searching for the minimum error between the experimental load–unload experimental data and the response of the models, with the PSO method, the values of the model parameters were determined. The responses of these adjusted models were compared with the experimental data from the staggered creep experiments, the anomalous values and the different behaviors were rejected, so their calculations were repeated. Finally, the models with the values of the parameters that managed to adequately follow the experimental behavior were accepted and their analysis proceeded.

### 4.1. Testing Samples

The studied material is an epoxy-matrix composite with random long vegetable fibers, see Figure 3a, described in [38]. The used polymer was PoliResinNovolac, provided by Polinova Company, Rio de Janeiro, Brazil. This thermosetting possesses good mechanical properties and resistance to soft acids, alkalis, and solar light. The novolac groups tend to increase the coupling with vegetable fibers [39], and the triethylenediamine-based curing agent allows the polymerization process at ambient temperature, avoiding fiber degradation. As reinforcement, the Cuban henequen plant fiber with an average length of 87 mm, and an average length/diameter ratio of 289.4, was used without any surface treatment. The experimental samples were cut from handmade plates, Figure 3b made in cold-closed molds, by weight percentage fiber/matrix of 0 wt%, 9 wt%, 14 wt%, 22 wt% and 28 wt%, see Figure 3c. All samples were tensile specimens from the ASTM D638-2014 norm type I.

### 4.2. Experimental Procedure

The standard monotonic tensile tests do not allow the acquisition of sufficient data to establish an accurate model parameter value. On the other hand, the process cost of the information is directly proportional to the complexity and data quantity. The present work intended to establish a balance between simplicity and quality in the obtained information.

All tests were conducted on a standard universal traction machine Instron 5567 with a maximum force of 30 kN, at room temperature 25 °C and 60% humidity. The strain and stress levels were selected to ensure non-breakage of the specimens, taking into account the results of a previous work [38]. The strain and stress data values were taken from the experimental points, that were measured every 0.5 s.

The cyclic load–unload tensile tests with 0.1%, 0.2%, 0.3%…1.0% of controlled strain levels (at 5 mm/min of strain rate), as a simple example see Figure 4, and the staggered fluency tests with 5 MPa, 10 MPa and 15 MPa of constant tension by stage (each maintained for 420 s), was made for five samples of each composite fiber/matrix percentage, as some simple examples see Figure 5 and Figure 6. It is necessary to point out in the loading–unloading test, at the end of each stage and change of direction some points were below zero, possibly caused by the machine inertia.

For the cyclic loading–unloading traction tests, five additional tests were performed and used for the validation stage and comparison between models, examples are provided in Figures 7–9. Similarly, the objectives of the fluency test data were the comparison of the adjusted model capacity to simulate the behavior of the studied material, validating the calculated constants, and a comparison between the fractional and the integer index models, Figures 11–13.

For each composite with different fiber/matrix percentages, initially making preliminary calculus using data only from the first cycles of the loading–unloading experimental points as entrance, with method constants adjustment as the objective. After several repetitions, these values were determined for each case, taking into account the resulting mode and variance. Using these values for the fiber-inclusion percentage level and then repeating the calculation process 12 times, the parameters of the integer and fractional index models were calculated. Monitoring the evolution of the iterative calculation process, when a process produced anomalous results it was interrupted and restarted, the most probable cause being the premature convergence of the PSO method.

## 5. Results

The parameters of the studied composite materials were determined and are presented in Table 1 and Table 2, for the integer derivative index and fractional derivative index models. The recorded values are similar to those found in [40,41], in addition, the Modulus of Elasticity is close to those determined in [42,43] through experiments with similar materials. In both cases, there is a tendency to increase the parameter values by increasing the quantity of reinforcement. Some small decreases are observed for 9wt% and 28 wt%, similar to what occurs under constant stress over time [38]. The first is due to insufficient reinforcing action of the fibers and the second is due to the critical fiber/matrix ratio being exceeded. In the case of the fractional index model, Table 2, the value of the derivative index tends to be one due to the increased influence of the viscous aspect of the fiber behavior.

The calculated parameters were validated through comparison with the experimental data. As examples, with fewer represented experimental points Figure 7, Figure 8 and Figure 9 show that the fractional index model can more accurately follow the behavior of the composite materials under loading–unloading cyclic tests. The error in both models concerning the experimental values increases as the deformation increases, reaching a maximum in the changes from load to unload. This is because as the stress (and strain) increases, the viscous component becomes more evident, and also the fact that the fractional index model better incorporates the influence of “strain history” or “memory effect”.

Taking the maximum values of each cycle, it can be observed in Figure 10 the percentage relative error between the responses of the integer index and fractional index models concerning the experimental data. From this, for the 9 wt%, 14 wt%, and 22 wt% of fiber inclusion, a decrease in the dispersion of the results. For all the reinforcement proportions studied, except for 28 wt%, a smaller error occurs fundamentally for the higher cycles in the response of the fractional index model. This anomaly occurs, presumably, because the critical volume of fiber addition was exceeded, so the matrix stops working cohesively, causing the composite material to malfunction.

Several comparisons were made between the integer index model and fractional index model with the staggered fluency test data, Figure 11, Figure 12 and Figure 13. In the first part of each stage, elastic deformation, both models and the experimental data behave in the same way, but this is not the case when the applied force is established as constant. It can be observed that the integer index model is incapable of changing the curve form on the second and third steps of the stress change, and therefore cannot follow the behavior of the studied material. On the other hand, the fractional index model changes the shape of the curve and improves the precision concerning the experimental data through the evolution of time, so that, with more information (experimental data) this model increases its capacity to simulate the behavior of the studied materials. The shape of the curves is conditioned, in materials by their “strain history” [44] and in models by the index of their derivatives [34].

The influence of the fiber/matrix ratio should be noted, mainly in the third stage, over the values (slight increase) and the curve form (slightly flatter for intermediate amounts of reinforcement inclusion). Two possible reasons are the restriction of fluency of the matrix and the increase in the viscous component provided by the vegetable-reinforcing fibers. Another fact is that the integer order model cannot handle a creep strain that is not zero at the beginning of the second and third stages, while the fractional index model can. This is due to the flexible change granted by the constitutive parameter α.

In Figure 14, the percentage error between the mean of the experimental values and the response of both models can be observed. The biggest errors occur in the first stage or stress level for both models, measuring the largest in materials with 14 wt% and 28 wt%, but in all cases, the values improve in the two following stages. In general, a decrease in the dispersion of the results occurs as the experiments progress. Excluding 0 wt%, in the second stage, the fractional index model presents predominantly lower error values concerning the experimental data. This greater accuracy is accentuated in the third stage across all cases, since the greater amount of information (experimental points) has more influence on the fractional index model than on the integer index model.

In future works, it would be advisable to extend the study to two-dimensional and three-dimensional configurations, as well as to investigate composite materials with other matrices and other reinforcing vegetable fibers. If greater precision of the results and a study of more complex materials with greater complexity regarding composition and behavior are necessary, more sophisticated experiments could be employed (with more stages, changes, and the combination of different forms of charge), including fractional index models with more elements and the use of more versatile numerical optimization methods (such as hybrids between several strategies).

## 6. Conclusions

The present work is mainly motivated by the need to acquire parameter values that adapt the behavior of the selected integer index and fractional index models to the evolution of study composites. Employing this study as a basic foundation, a one-dimensional model can be developed and generalized to a two-dimensional and three-dimensional version.

The experimental data and the model results indicate a strong dependence between the material properties and the percentage of the reinforcement-fibers included, resulting in significant value differences.

Both models demonstrated the capability of simulating long-term material behavior under cyclic loading–unloading and staggered fluency conditions. However, on the other hand, the fractional index model was shown to follow the form of the experimental curves with greater accuracy in both tests.

Regarding the staggered fluency test graphs, the fractional index model adjustment is shown to produce better results in the third step than in the first experimental step. This model can modify the shape of the curve for different load conditions and their changes over time. Its accuracy in modeling material mechanical behavior improves as the amount of experimental data increases.

## Figures and Tables

**Figure 1 polymers-14-04634-f001:**
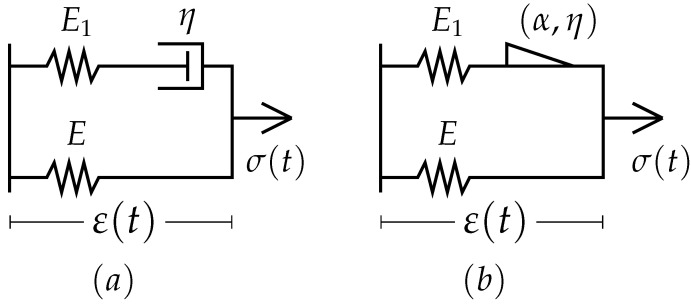
(**a**) Standard model or Linear Viscoelastic Solid. (**b**) The fractional version or fractional Zener Model.

**Figure 2 polymers-14-04634-f002:**
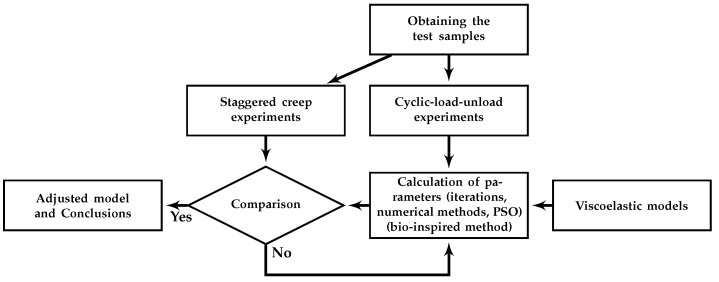
General scheme of the work.

**Figure 3 polymers-14-04634-f003:**
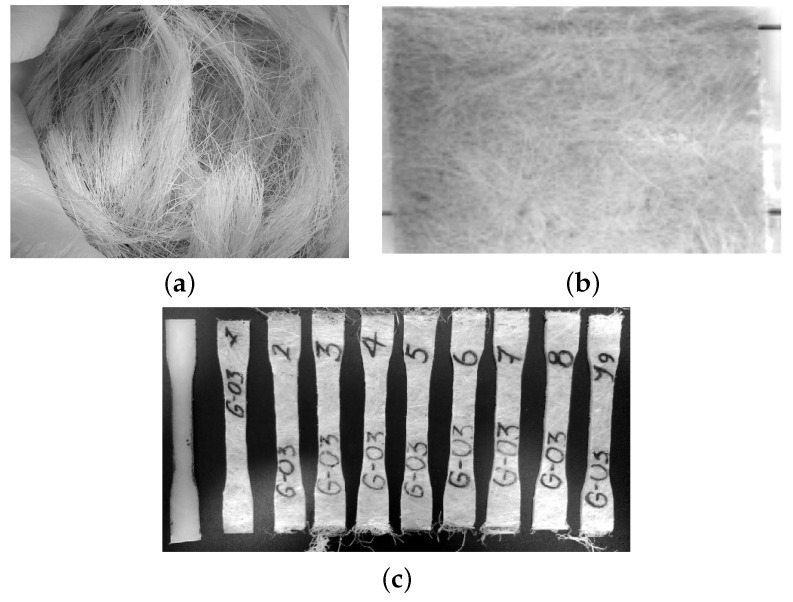
(**a**) Dry henequen fibers. (**b**) Composite plate. (**c**) Typical traction samples of fiber-reinforced composites.

**Figure 4 polymers-14-04634-f004:**
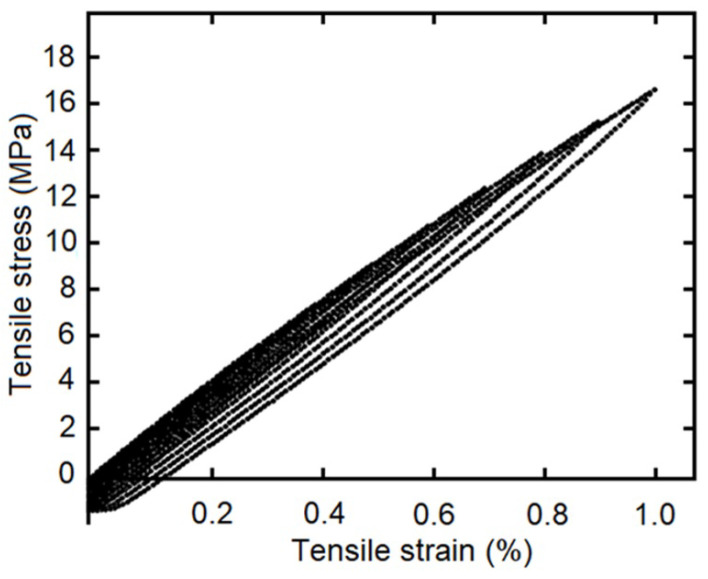
Loading–unloading tensile tests with strain as entrance signal, one specimen with 9 wt%.

**Figure 5 polymers-14-04634-f005:**
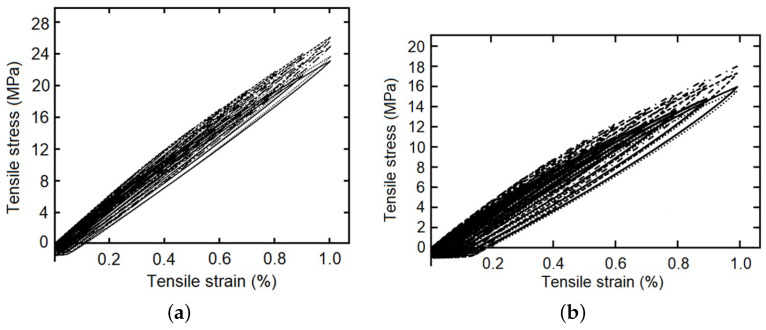
Loading–unloading tensile tests with strain as entrance signal, five specimens, (**a**) with 22 wt%, and (**b**) with 28 wt%.

**Figure 6 polymers-14-04634-f006:**
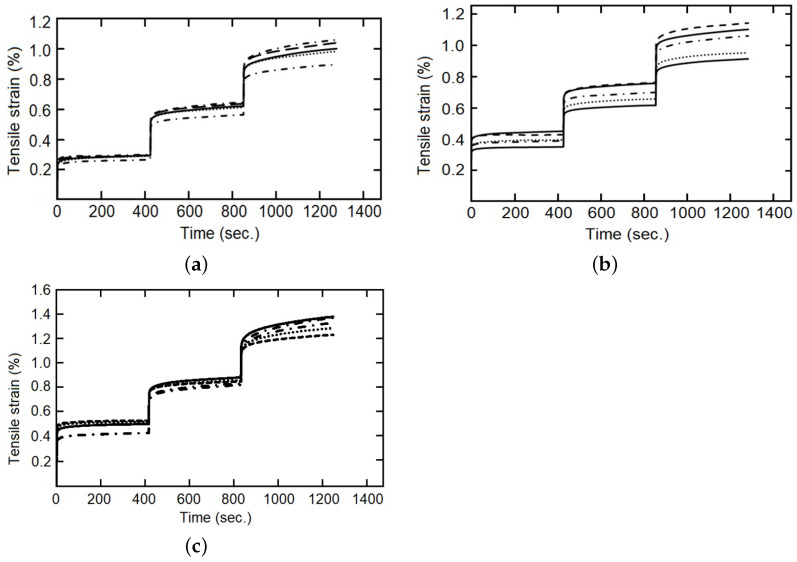
Fluency staggered test, five specimens, (**a**) with 9 wt%, (**b**) with 14 wt%, and (**c**) with 28 wt%.

**Figure 7 polymers-14-04634-f007:**
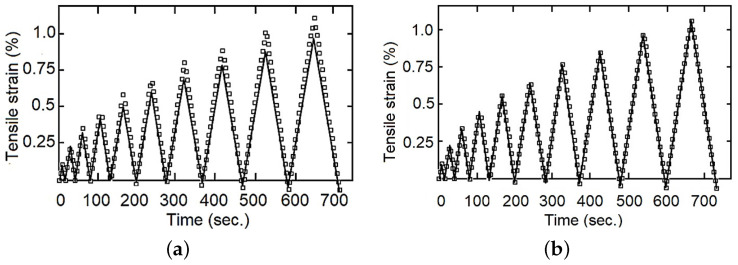
Tensile strain for loading–unloading test, one specimen with 0 wt%, experimental data in dots and model response in continuous line, (**a**) integer index model, (**b**) fractional index model.

**Figure 8 polymers-14-04634-f008:**
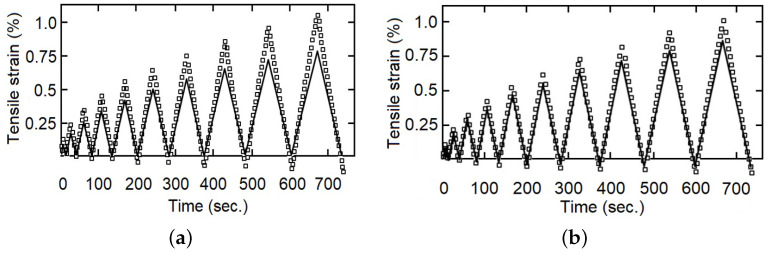
Tensile strain for loading–unloading test, one specimen with 9 wt%, experimental data in dots and model response in continuous line, (**a**) integer index model, (**b**) fractional index model.

**Figure 9 polymers-14-04634-f009:**
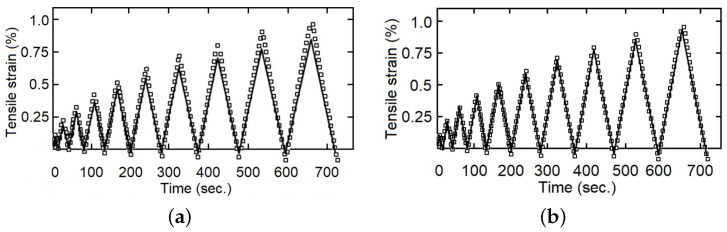
Tensile strain for loading–unloading test, one specimen with 14 wt%, experimental data in dots and model response in continuous line, (**a**) integer index model, (**b**) fractional index model.

**Figure 10 polymers-14-04634-f010:**
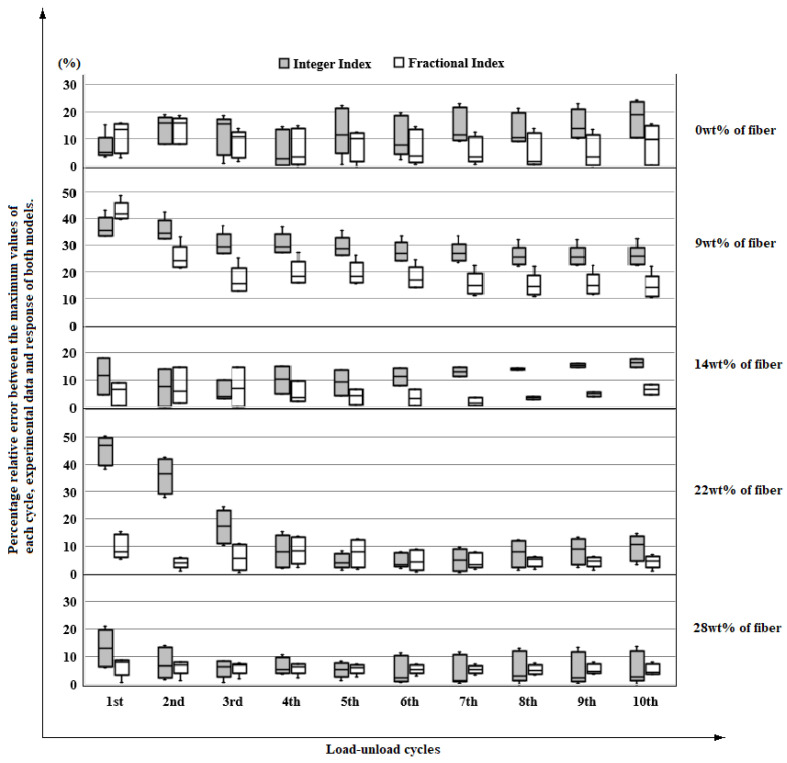
Percentage relative error between the maximum values of each cycle, experimental data and response of both models.

**Figure 11 polymers-14-04634-f011:**
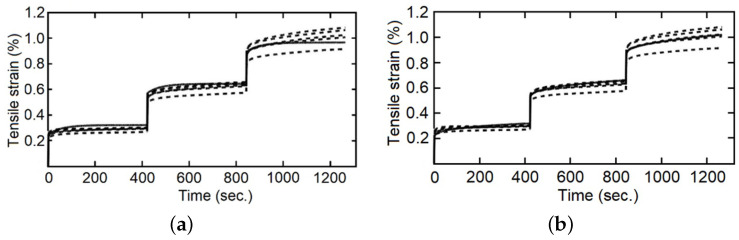
Tensile strain for staggered fluency test, five specimens with 9 wt%, experimental data in dotted lines and model response in continuous line, (**a**) integer index model, (**b**) fractional index model.

**Figure 12 polymers-14-04634-f012:**
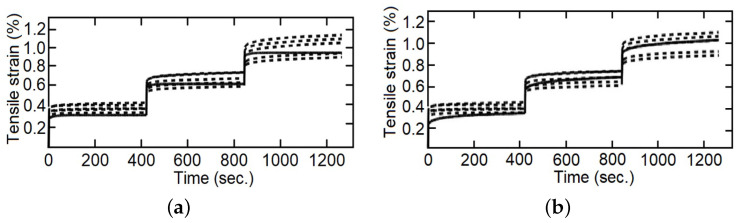
Tensile strain for staggered fluency test, five specimens with 14 wt%, experimental data in dotted lines and model response in continuous line, (**a**) integer index model, (**b**) fractional index model.

**Figure 13 polymers-14-04634-f013:**
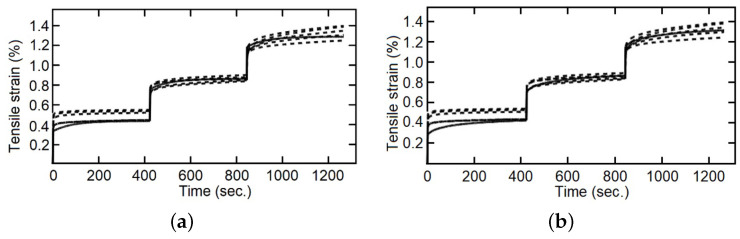
Tensile strain for staggered fluency test, five specimens with 28 wt%, experimental data in dotted lines and model response in continuous line, (**a**) integer index model, (**b**) fractional index model.

**Figure 14 polymers-14-04634-f014:**
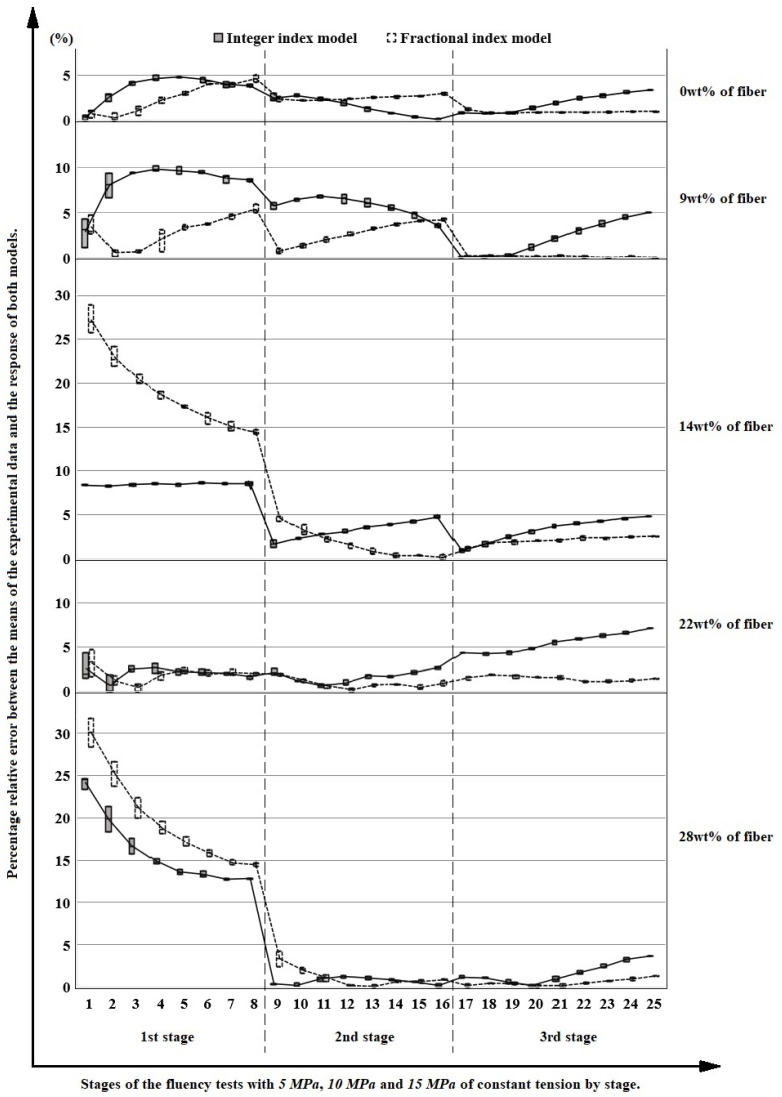
Percentage relative error between the means of the experimental data and the response of both models, integer index model in a continuous line and gray fill, and fractional index model in dashed line and white fill.

**Table 1 polymers-14-04634-t001:** Parameters of the integer model.

Fiber Percent in Weight	η (GPa· s)	$E_{1}$ (GPa)	*E* (GPa)	α
0 wt%	10.4429	1.8643	0.3007	1
9 wt%	15.0150	2.3142	0.6222	1
14 wt%	15.6175	2.3624	0.6427	1
22 wt%	16.3209	2.5003	0.6512	1
28 wt%	14.9841	2.2026	0.5986	1

**Table 2 polymers-14-04634-t002:** Parameters of the Zener fractional index model.

Fiber Percent in Weight	η (GPa· s)	$E_{1}$ (GPa)	*E* (GPa)	α
0 wt%	9.9695	0.5874	0.9964	0.2155
9 wt%	8.1984	1.6223	0.5998	0.4993
14 wt%	10.0255	2.1003	0.6069	0.7996
22 wt%	12.0033	2.3232	1.1035	0.8101
28 wt%	16.0550	2.1599	1.8237	0.6713

## Data Availability

Not applicable.

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
