# Peer review of "Parameter Identification of Fractional Index Viscoelastic Model for Vegetable-Fiber Reinforced Composite"

_polymers, 2022, doi:10.3390/polym14214634_

Round 1

Reviewer 1 Report

Review of parameter identification of fractional index viscoelastic model for vegetable fibre reinforced composite. The following are my comments and suggestions

·         The aim, motivation and the significant of the study should be clearly stated in the last paragraph of the introduction section.

·         What is new in the study? Kindly indicate the novelty of the study.

·         How tendency to agglomeration of material was taken into account in the presented model? (Hint: update formulation of the problem section assumptions).

·         Please clarify the application of this investigation.

·         Some grammatical mistakes should be checked through.

·         All figures should be discussed from the physical implications, significance justification point of view, hence, the discussion of results section needs to be improved.

·         Why has the author chosen to engage in this study?

·         Likely extension of the study should be stated or recommended. 

Good work

Author Response

Good morning, I hope you are well.

In this letter, we detail the changes made to the article. Based on the suggestions made by the reviewers, all modifications are marked in red.

Thanking you for your comments and guidance. The topics discussed follow:

  • The aim, motivation, and significance of the study should be clearly stated in the last paragraph of the introduction section.
  • What is new in the study? Kindly indicate the novelty of the study.

Both points have been included in the last paragraph (lines 89-104) of the introduction, leaving this modified:

“Nowadays, there are no models adjusted to the mechanical behavior of thermosetting polymeric matrix composite materials reinforced with long random fibers of henequen, sisal, or similar. Motivated by the need to be able to simulate the behavior over time of these materials under the action of external loads, the main objective of this work is to calculate the values of the parameters of the models that allow it. For this, we propose an innovative approach, selecting a rheological model with fractional index derivatives to address the viscoelastic behavior of these polymeric matrix compounds reinforced with vegetable fibers. Hence a smaller amount of experimental data is necessary for the adjustment of the model, compared to more complex models with integer index derivatives. Also, it is necessary to highlight that the present work proposes a methodology that tries to maintain simplicity and low computational cost while ensuring the simulation capacity of the used model. For this reason, was used the Particle Swarm Optimization (PSO) method to the resolution of the Inverse Identification problem, through the minimization of the error between the experimental data and those obtained by the models. For the calculation and the validation of the parameters was use experimental data from simple cyclic incrementing loading-unloading tests and stepped fluency tests of brief time duration.”

  • How tendency to agglomeration of material was taken into account in the presented model? (Hint: update formulation of the problem section assumptions).

We modified the last paragraph (lines 145-148) of the "Material model" subsection, including the following considerations:

“Also, used the effective global properties of the composite as an equivalent homogeneous material [33], and assumed a uniform distribution and a random orientation of the reinforcing fibers. In addition, are not consider the possible agglomerations of the reinforcements, allowing for assuming global isotropic properties in the plane of the loads.”

  • Please clarify the application of this investigation.

We modified the second paragraph (lines 27-34) of the introduction, leaving this modified:

“For the design of parts and pieces made of these materials, it is desirable to have digital models that can simulate their behavior over time under the action of external loads. These models, adjusted to the real behavior of the composites and having the ability to consider their "strain history", can be implemented, for example, in Finite Element Method (FEM) analysis programs. As a basic first step, the unidimensional models can be generalized to two-dimensional and three-dimensional simulations. Therefore, it is necessary to determine the constants which adapt these models to the behavior of the composite material, possible only by the analysis of experimental data.”

  • Some grammatical mistakes should be checked through:

The text was revised, and several grammatical errors were corrected.

  • All figures should be discussed from the physical implications, and significance justification point of view, hence, the discussion of the results section needs to be improved.

We modified the first paragraph (lines 258-268) of the results section, leaving this modified:

“The parameters of the composite materials studied have been determined, which are presented in Tables 1 and 2, for the integer derivative index and fractional derivative index models. The recorded values are similar to those found in [38] and [40], in addition, the Modulus of Elasticity is close to those determined in [39] and [41] through experiments with similar materials. In both cases, there is a tendency to increase the parameter values by increasing the quantity of reinforcement. Some small declines are observed for 9wt% and 28wt%, similar to what happens under constant stress over time [42]. The first is due to the insufficient reinforcing action of the fibers and the second is because the critical fiber/matrix ratio is being exceeded. In the case of the fractional index model, Table 2, the value of the derivative index tends to be one due to the increased influence of the viscous part of the fiber behavior.”

We modified the second paragraph (lines 269-276) of the results section, leaving this modified:

“The validation of the calculated parameters was made through comparison with the experimental data. As examples, with fewer represented experimental points Figures 7 to 9 show that the fractional index model can follow more accurately the behavior of the composite materials under load-unload cyclic tests. The error of both models concerning the experimental values increases with the increase of the deformation, reaching its maximum in the changes from load to unload. This is because as the stress (and strain) increases, the viscous component becomes more evident. In addition to the fact that the fractional index model better incorporates the influence of "strain history" or "memory effect".”

We modified the third paragraph (lines 277-284) of the results section, leaving this modified:

“Taking the maximum values of each cycle, it can be observed in Figure 10 the percentage relative error between the responses of the integer index and fractional index models concerning the experimental data. From this, for the 9wt%, 14wt%, and 22wt% of fiber inclusion, a decrease in the dispersion of the results. For all the reinforcement proportions studied, except for 28wt%, a smaller error occurs fundamentally for the higher cycles in the response of the fractional index model. This anomaly occurs, presumably, because the critical volume of fiber addition was exceeded, so the matrix stops working cohesively, causing the composite material to malfunction.”

We modified the fourth paragraph (lines 285-295) of the results section, leaving this modified:

“Made several comparisons between integer index model and fractional index model with the staggered fluency test data, Figures from 11 to 13. In the first part of each stage, elastic deformation, both models and the experimental data behave in the same way, but this is not the case when the applied force is established as constant. Can be observed that the integer index model is not capable to change the curve form on the second and third steps of the stress change, therefore, it can’t follow the behavior of the studied material. On the other hand, the fractional index model changes the shape of the curve and improves the precision concerning the experimental data through the evolution of time, so that, with more information (experimental data) this model increases its capacity to simulate the behavior of the studied materials. The shape of the curves is conditioned, in materials by their "strain history" [44] and in models by the index of their derivatives [34].”

We modified the fifth paragraph (lines 296-302) of the results section, leaving this modified:

“An influence of the fiber/matrix ratio should be noted, mainly in the third stage, over the values (slight increase) and the curve form (slightly flatter for intermediate amounts of reinforcement inclusion). The two possible reasons are the restriction of the fluency of the matrix and the increase in the viscous component provided by the vegetable reinforcing fibers. Another fact is that the integer order model cannot handle that the creep strain is not zero at the beginning of the second and third stages, while the fractional index model can. This is due to the flexible change that the constitutive parameter α grants.”

We modified the sixth paragraph (lines 303-311) of the results section, leaving this modified:

“In Figure 14 can be observed the percentage error between the mean of the experimental values and the response of both models. The biggest errors occur in the first stage or stress level for both models, being the largest in the materials with 14wt% and 28wt%, but in all cases, the values improve in the two following stages. In general, a decrease in the dispersion of the results occurs with the progress of the experiments. Except in 0wt%, in the second stage, the fractional index model presents predominantly lower error values concerning the experimental data. This greater accuracy is accentuated in the third stage in all cases, this is because the greater amount of information (experimental points) has more influence on the fractional index model than on the integer index model.”

  • Why has the author chosen to engage in this study?

To design parts made of these or similar materials, it is necessary to have models that can simulate their mechanical behavior. In the bibliography consulted up to now, there are no models of viscoelastic behavior adapted for these composites (with a polymeric matrix reinforced with vegetable fibers). These models can be implemented, for example, in calculation programs using FEM and provide information on the parts under the action of different loads over time. This first approximation of the uniaxial models can later be converted to 2D and 3D versions. To contribute to the speed of the engineering design processes, a simple methodology, non-complex experiments, validation through graphical comparison, and the use of numerical methods to obtain the results are proposed.

  • Likely extension of the study should be stated or recommended.

We modified the last paragraph (lines 312-319) of the results section, leaving this modified:

“In future works, it would be advisable to extend the study to two-dimensional and three-dimensional configurations, as well as to investigate composite materials with other matrices and other reinforcing vegetable fibers. If greater precision of the results and the study of materials with greater complexity in their composition and behavior are necessary, more complex experiments could be used (with more stages, changes, and the combination of different forms of charge), including models fractional index with more elements and use more versatile numerical optimization methods (such as hybrids between several strategies).”

Any other comments, questions, or information needed, please, do not hesitate to let us know.

Thanking you for your comments and guidance.

Best regards.

Reviewer 2 Report

·             The research problem in the paper does not seem to be motivated by a clearly outlined research question and no physical insight is provided for this theoretical analysis. So, the authors need to address such deficiencies. I can support publication after the authors undertake this point clearly

·              

·              In the results section, the author needs to analyse the finding by giving reasons for each fact. Please explain every point?

·             The paper should be overviewed against the grammatical error.

·             For more contribution, the authors should compare their results with the related results in other published works such as

·             The range of considered variables must be put in the Abstract.

·             The originality points and the practical applications of this work must be added

·             The practical application of the present study must be highlighted

·             Proofreading by a native English speaker or proofreading service should be conducted to improve both language and organization quality.

·             Please modify the title of the paper as it does not well reflect the work carried out.

·             The section "Abstract" is too short and insufficient. Please improve this section.

·             The novelty and originality of the work have not been well explained. The authors might add a Table to the section "Introduction" to summarize the research papers published on this subject to address this issue

Physica A: Statistical Mechanics and its Applications 525, 741-751, 2019,
Physica A: Statistical Mechanics and Its Applications 525, 616-627, 2019 Journal of Environmental Health Science and Engineering 17 (1), 53-62, 2019 Physica A: Statistical Mechanics and its Applications 546, 123995, 2020

Author Response

Good morning, I hope you are well.

In this letter, we detail the changes made to the article. Based on the suggestions made by the reviewers, all modifications are marked in red.

Thanking you for your comments and guidance. The topics discussed follow:

  • The research problem in the paper does not seem to be motivated by a clearly outlined research question and no physical insight is provided for this theoretical analysis. So, the authors need to address such deficiencies. I can support publication after the authors undertake this point clearly.
  • In the results section, the author needs to analyze the finding by giving reasons for each fact. Please explain every point.
  • The originality points and the practical applications of this work must be added.
  • The practical application of the present study must be highlighted.

Concerning these points:

We modified the second paragraph (lines 27-34) of the introduction, leaving this modified:

“For the design of parts and pieces made of these materials, it is desirable to have digital models that can simulate their behavior over time under the action of external loads. These models, adjusted to the real behavior of the composites and having the ability to consider their "strain history", can be implemented, for example, in Finite Element Method (FEM) analysis programs. As a basic first step, the unidimensional models can be generalized to two-dimensional and three-dimensional simulations. Therefore, it is necessary to determine the constants which adapt these models to the behavior of the composite material, possible only by the analysis of experimental data.”

We modified the last paragraph (lines 89-104) of the introduction, leaving this modified:

“Nowadays, there are no models adjusted to the mechanical behavior of thermosetting polymeric matrix composite materials reinforced with long random fibers of henequen, sisal, or similar. Motivated by the need to be able to simulate the behavior over time of these materials under the action of external loads, the main objective of this work is to calculate the values of the parameters of the models that allow it. For this, we propose an innovative approach, selecting a rheological model with fractional index derivatives to address the viscoelastic behavior of these polymeric matrix compounds reinforced with vegetable fibers. Hence a smaller amount of experimental data is necessary for the adjustment of the model, compared to more complex models with integer index derivatives. Also, it is necessary to highlight that the present work proposes a methodology that tries to maintain simplicity and low computational cost while ensuring the simulation capacity of the used model. For this reason, was used the Particle Swarm Optimization (PSO) method to the resolution of the Inverse Identification problem, through the minimization of the error between the experimental data and those obtained by the models. For the calculation and the validation of the parameters was use experimental data from simple cyclic incrementing loading-unloading tests and stepped fluency tests of brief time duration.”

We modified the second paragraph (lines 269-276) of the results section, leaving this modified:

“The validation of the calculated parameters was made through comparison with the experimental data. As examples, with fewer represented experimental points Figures 7 to 9 show that the fractional index model can follow more accurately the behavior of the composite materials under load-unload cyclic tests. The error of both models concerning the experimental values increases with the increase of the deformation, reaching its maximum in the changes from load to unload. This is because as the stress (and strain) increases, the viscous component becomes more evident. In addition to the fact that the fractional index model better incorporates the influence of "strain history" or "memory effect".”

We modified the third paragraph (lines 277-284) of the results section, leaving this modified:

“Taking the maximum values of each cycle, it can be observed in Figure 10 the percentage relative error between the responses of the integer index and fractional index models concerning the experimental data. From this, for the 9wt%, 14wt%, and 22wt% of fiber inclusion, a decrease in the dispersion of the results. For all the reinforcement proportions studied, except for 28wt%, a smaller error occurs fundamentally for the higher cycles in the response of the fractional index model. This anomaly occurs, presumably, because the critical volume of fiber addition was exceeded, so the matrix stops working cohesively, causing the composite material to malfunction.”

We modified the fourth paragraph (lines 285-295) of the results section, leaving this modified:

“Made several comparisons between integer index model and fractional index model with the staggered fluency test data, Figures from 11 to 13. In the first part of each stage, elastic deformation, both models and the experimental data behave in the same way, but this is not the case when the applied force is established as constant. Can be observed that the integer index model is not capable to change the curve form on the second and third steps of the stress change, therefore, it can’t follow the behavior of the studied material. On the other hand, the fractional index model changes the shape of the curve and improves the precision concerning the experimental data through the evolution of time, so that, with more information (experimental data) this model increases its capacity to simulate the behavior of the studied materials. The shape of the curves is conditioned, in materials by their "strain history" [44] and in models by the index of their derivatives [34].”

We modified the fifth paragraph (lines 296-302) of the results section, leaving this modified:

“An influence of the fiber/matrix ratio should be noted, mainly in the third stage, over the values (slight increase) and the curve form (slightly flatter for intermediate amounts of reinforcement inclusion). The two possible reasons are the restriction of the fluency of the matrix and the increase in the viscous component provided by the vegetable reinforcing fibers. Another fact is that the integer order model cannot handle that the creep strain is not zero at the beginning of the second and third stages, while the fractional index model can. This is due to the flexible change that the constitutive parameter α grants.”

We modified the sixth paragraph (lines 303-311) of the results section, leaving this modified:

“In Figure 14 can be observed the percentage error between the mean of the experimental values and the response of both models. The biggest errors occur in the first stage or stress level for both models, being the largest in the materials with 14wt% and 28wt%, but in all cases, the values improve in the two following stages. In general, a decrease in the dispersion of the results occurs with the progress of the experiments. Except in 0wt%, in the second stage, the fractional index model presents predominantly lower error values concerning the experimental data. This greater accuracy is accentuated in the third stage in all cases, this is because the greater amount of information (experimental points) has more influence on the fractional index model than on the integer index model.”

  • The paper should be overviewed against grammatical errors.

The text was revised, and several grammatical errors were corrected.

  • For more contribution, the authors should compare their results with the related results in other published works such as.

We modified the first paragraph (lines 258-268) of the results section, leaving this modified:

“The parameters of the composite materials studied have been determined, which are presented in Tables 1 and 2, for the integer derivative index and fractional derivative index models. The recorded values are similar to those found in [38] and [40], in addition, the Modulus of Elasticity is close to those determined in [39] and [41] through experiments with similar materials. In both cases, there is a tendency to increase the parameter values by increasing the quantity of reinforcement. Some small declines are observed for 9wt% and 28wt%, similar to what happens under constant stress over time [42]. The first is due to the insufficient reinforcing action of the fibers and the second is because the critical fiber/matrix ratio is being exceeded. In the case of the fractional index model, Table 2, the value of the derivative index tends to be one due to the increased influence of the viscous part of the fiber behavior..”

  • The range of considered variables must be put in the Abstract.

We modified the abstract:

In the present work, the parameters that adapt the behavior of the uniaxial three-element viscoelastic constitutive model with integer and fractional index derivatives to the mechanical evolution of the epoxy composite material reinforced with long random fibers of Henequen were determined. Cyclic-load-unload with 0.1%, 0.2%,0.3%, . . . ,1.0% of controlled strain and staggered fluency experiments at 5MPa,10MPa, and 15MPa of constant tension by stage were performed, and the obtained data were used to determine the model’s parameter values and its validations. The calculation of the parameters used the Inverse Method of Identification, and the minimization of the error function was made using the Particle Swarm Optimization (PSO) method. The comparison between the simulated uniaxial results and the experimental data is shown through graphs. Exist a strong dependence between the properties of the composite and the fiber content (0wt%, 9wt%, 14wt%, 22wt%, and 28wt% of weight percentages fiber/matrix), and therefore also of the values of the model parameters. Both uniaxial models can follow the viscoelastic behavior of the material and the fractional index version presents the best accuracy. Was appreciated that the following method is adequate to determine these kinds of constants using non-large experimental data and procedures easy to implement.

Any other comments, questions, or information needed, please, do not hesitate to let us know.

Thanking you for your comments and guidance.

Best regards.
